# Development of highly sensitive and low-cost DNA agarose gel electrophoresis detection systems, and evaluation of non-mutagenic and loading dye-type DNA-staining reagents

**Ken Motohashi**[1,2]*

**1** Department of Frontier Life Sciences, Faculty of Life Sciences, Kyoto Sangyo University, Kamigamo Motoyama, Kita-ku, Kyoto, Japan, **2** Center for Ecological Evolutionary Developmental Biology, Kyoto Sangyo University, Kamigamo Motoyama, Kita-Ku, Kyoto, Japan

* motohas@cc.kyoto-su.ac.jp

**Data Availability Statement:** All relevant data are within the manuscript and its Supporting Information files.

## Abstract

Highly sensitive and low-cost DNA agarose gel detection systems were developed using non-mutagenic and loading dye-type DNA-staining reagents. The DNA detection system that used Midori Green Direct and Safelook Load-Green, both with an optimum excitation wavelength at ~490 nm, could detect DNA-fragments at the same sensitivity to that of the UV (312 nm)-transilluminator system combined with ethidium bromide, after it was excited by a combination of cyan LED light and a shortpass filter (510 nm). The cyan LED system can be also applied to SYBR Safe that is widely used as a non-toxic dye for post-DNA-staining. Another DNA-detection system excited by black light was also developed. Black light used in this system had a peak emission at 360 nm and caused less damage to DNA due to lower energy of UV rays with longer wavelength when compared to those of short UV rays. Moreover, hardware costs of the black light system were ~$100, less than 1/10 of the commercially available UV (365 nm) transilluminator (>$1,000). EZ-Vision and Safelook Load-White can be used as non-mutagenic and loading dye-type DNA-staining reagents in this system. The black light system had a greater detection sensitivity for DNA fragments stained by EZ-Vision and Safelook Load-White compared with the commercially available imaging system using UV (365 nm) transilluminator.

## Introduction

DNA separation and detection by agarose gel electrophoresis is one of the most frequently used techniques in life sciences [1–3]. Traditionally, DNA fragments loaded on agarose gels have been stained with ethidium bromide and detected by ultraviolet (UV)-transilluminator system [1, 4–7]. This system is a highly sensitive and low-running-cost method that has been used by many molecular biology researchers to visualize DNA in agarose gel after electrophoresis [8–11]. However, ethidium bromide and UV-light system require careful handling of the staining solution and short wavelength UV (312 nm)-transilluminator because of the

**Funding:** This work was supported by JSPS KAKENHI Grant Numbers 16K07409 (to K.M.), the MEXT-Supported Program for the Strategic Research Foundation at Private Universities Grant Number S1511023 (to K.M.), and the Institute for Fermentation (Osaka) Grant Number G-2019-2-067 (to K.M.). The funders had no role in study design, data collection and analysis, decision to publish, or preparation of the manuscript.

**Competing interests:** The author has declared that no competing interests exist.

mutagenic effects of ethidium bromide and high energy of short-wavelength UV light that are harmful to users [12, 13]. Moreover, DNA strands are damaged by high-energy short-wave UV rays [14]. Non-mutagenic alternative DNA-staining reagents, such as SYBR-Green and SYBR-Gold, have been developed to eliminate the disadvantages of DNA staining with ethidium bromide [15–20].

Recently, non-mutagenic and loading dye-type DNA-staining reagents that are simply mixed with sample DNA solutions have been developed and made available by several suppliers, such as EZ-vision (VWR Life Science, Radnor, PA, USA), Midori Green Direct (Nippon Genetics, Tokyo, Japan), Novel Juice (Bio-Helix, Keelung City, Taiwan) and Safelook-series (Fujifilm Wako Pure Chemical Corporation, Osaka, Japan). These DNA-staining reagents are used in small amounts, thereby bearing low cost when compared with other post-staining types or precast-gel type reagents such as SYBR series, and, owing to their low toxicity, they can be used safely to detect DNA fragments on agarose gels after electrophoresis [21–26]. However, the use of these novel non-mutagenic DNA-staining reagents requires an additional excitation light and optical filter system to acquire high detection sensitivity for DNA fragments; these reagents are excited by wavelengths that are longer than the short wavelengths of the UV light used with ethidium bromide DNA staining. Moreover, detection sensitivity of these non-mutagenic DNA-staining reagents is lower than that of ethidium bromide-UV transilluminator system, and excitation by longer UV wavelengths employed with those novel reagents causes less damage to DNA strands [18]. Here, two novel, simple, and low-cost DNA detection systems, using non-mutagenic and loading dye-type of DNA-staining reagents, to detect DNA fragments by agarose gel electrophoresis were developed. In addition, sensitivities of loading dye-type DNA-staining reagents were evaluated in optimized DNA-detection systems using the excitation-light systems.

## Materials and methods

### Materials

Midori Green Direct (Nippon Genetics), Novel Juice (Bio-Helix), Safelook Load-Green (Fujifilm Wako Pure Chemical Corporation), EZ-Vision One (VWR Life Science), and Safelook Load-White (Fujifilm Wako Pure Chemical Corporation) were used as loading dye-type of DNA-staining reagents. Midori Green Direct was supplied as 10× loading dye. Novel Juice, Safelook Load-Green, EZ-Vision One, and Safelook Load-White were supplied as 6× loading dye. SYBR Safe (Thermo Fisher Scientific, Carlsbad, CA) was used as a major nontoxic post-DNA-staining reagent. A 1Kb DNA Ladder RTU (Bio-Helix, 100 ng/μL) was used as DNA ladder marker (10, 8.0, 6.0, 5.0, 4.0, 3.0, 2.5, 2.0, 1.5, 1.0, 0.75, 0.50, 0.25 kbp). Agarose S (Nippon Gene, Tokyo, Japan) was used for DNA gel electrophoresis.

### DNA agarose gel electrophoresis

DNA agarose gel electrophoresis was performed for 25 min at 150 V with 0.8% agarose gel in a buffer containing 89 mM Tris, 89 mM borate, and 2 mM EDTA (TBE) [1, 7].

### Excitation light and filter system

Compact LED viewer (https://www.sanplatec.co.jp/product_pages.asp?arg_product_id=SAN26955, Sanplatec, Osaka, Japan) as blue LED light (470 nm) and cyan LED stand light (https://item.rakuten.co.jp/holkin-shop/hlk-12led-490nm-495nm/, hlk-12led-490nm-495nm; Holkin, Gifu, Japan) as cyan LED light (490–495 nm) were used to excite Midori Green Direct, Novel Juice, and Safelook Load-Green. Compact black light blue lamp 27 W (https://item.

rakuten.co.jp/denzaido/4992712060051/, 360 nm, FPL27BLB; Sankyo Denki, Tokyo, Japan) was used as black light to excite EZ-Vision and Safelook Load-White. A shortpass filter (https://www.asahi-spectra.co.jp/asp/syousaik.asp?key1=f41&key2=SV0510, 50 × 50 mm, SV0510; Asahi Spectra, Tokyo, Japan) was used as an excitation light filter for the cyan LED system.

### Emission filters

The common filters for photography, SC-42 (420 nm), SC-46 (460 nm), SC-48 (480 nm), SC-52 (520 nm), SC-54 (540 nm), and SC-56 (560 nm) (75 × 75 mm; Fujifilm, Tokyo, Japan) were used as longpass filters for emission systems.

### Commercially available imaging systems for DNA agarose gel electrophoresis

STAGE-One (AMZ System Science, Osaka, Japan) was used to visualize DNA ladder markers with ethidium bromide by UV-transilluminator system (312 nm excitation). STAGE-2000 (AMZ System Science) was used to visualize DNA ladder markers with EZ-vision and Safelook Load-White by UV-transilluminator system (365 nm excitation) as a control of commercially available UV-transilluminator system for EZ-vision and Safelook Load-White. Blue-LED transilluminator Blook$^{TM}$ (Bio-Helix) was used to visualize DNA ladder markers with Midori Green Direct as a control of commercially available blue-LED transilluminator system for Midori Green Direct.

### Photograph system

Images of DNA agarose gels after electrophoresis were recorded with a digital camera Power-Shot G12 (Canon, Tokyo, Japan) mounted on a photo stand shaded by blackout curtain. None of the images were cropped, and altered by image processing methods such as contrasting and enhancement.

### Determination of detection sensitivities by successive dilutions of DNA markers on agarose gels

Fluorescent intensity of successive dilutions of DNA markers on agarose gels was analyzed for each sample by "Plot Profile" of Image-J [27]. Detected sensitivity was defined by the dilution when all the peaks of 13 DNA marker fragments were detected. The detection limit at standard volume, 1/3–1/6 volume, and <1/10 volume of DNA markers was defined as moderate, moderate-high, and high, respectively. When a part of the 13 peaks could not be detected with standard volume (5 μL), the sensitivity was defined as low.

## Results and discussion

### Excitation and emission of non-mutagenic and loading dye-type DNA-staining reagents

Non-mutagenic and loading dye-type DNA-staining reagents used in this study were classified according to their excitation wavelength required for detection of DNA fragments in agarose gel after electrophoresis (Table 1). Midori Green Direct, Novel Juice, and Safelook Load-Green were excited by blue (470 nm) or cyan (490–495 nm) light and showed green fluorescence. EZ-Vision and Safelook Load-White were excited by ultraviolet-A (~365 nm) and they release blue fluorescence. In contrast, ethidium bromide was well-excited by short-wave UV rays (312

**Table 1. Excitation and emission wavelengths of non-mutagenic and loading dye-type DNA-staining reagents.**

| Excitation light | DNA-staining reagent[*1] | Excitation wavelength (nm)[*2] | Emission wavelength (nm)[*2] |
|---|---|---|---|
| blue or | Midori Green Direct | 490 | 530 |
| cyan LED | Novel Juice | 495 | 537 |
|  | Safelook Load-Green | 490 | 525 |
|  | SYBR Safe[*3] | 502 | 530 |
| UV-A | EZ-Vision | 364 | 454 |
|  | Safelook Load-White | 370 | 470 |

[*1]Midori Green Direct was supplied as 10× loading dye. Novel Juice, Safelook Load-Green, EZ-Vision, and Safelook Load-White were supplied as 6× loading dye.

[*2]Optimum excitation and emission wavelengths for DNA-staining reagents were described in each instruction manual.

[*3]SYBR Safe is listed because the reagent is widely used as a non-mutagenic DNA-staining reagent, although the reagent is of a post-staining type or precast-gel type.

nm) with high energy. Detection limit of the ethidium bromide staining was evaluated by successive dilutions of DNA markers (standard volume to 1/30 volume) (Fig 1A). The detection limit of staining reagents was defined by the dilution when all peaks of 13 DNA maker fragments were detected, as described in materials and methods (Fig 1 and S1 Fig). DNA markers stained with ethidium bromide could be detected by using 1/10 of standard DNA marker volume (500 ng, 5 μL). This result indicated high sensitivity of ethidium bromide DNA staining. In this study, two DNA detection systems for agarose gel electrophoresis were built by combining a common, commercially available LED or black light (S2 Fig) and optical filters for photography, using loading dye-type DNA-staining reagents. Schematic diagram of these DNA detection systems and photographs of the systems are shown in Fig 2 and S3 Fig, respectively.

## High-sensitivity detection system of DNA fragments by blue or cyan LED light-excited DNA-staining reagents

Midori Green Direct, Novel Juice, and Safelook Load-Green are generally excited by blue LED light system. Actually, commercially available Blook system that is equipped by blue LED (470 nm) could detect Midori Green Direct-stained DNA markers (Fig 1A). The sensitivity of Midori Green Direct-stained DNA markers was evaluated by low-cost blue LED light (Compact LED viewer [470 nm], Sanplatec) whose price was ~$180 (Fig 1B and S4A Fig). Midori Green Direct-stained DNA markers were also detected by the low-cost blue LED viewer when 1/3 volume of DNA markers was loaded on an agarose gel, although the sensitivity was slightly less than that of the Blook system, which could detect 1/6 volume of DNA markers (Fig 1A and 1B). Next, to develop a low-cost DNA detection system with higher sensitivity using non-mutagenic DNA-staining reagents, Cyan LED (490–495 nm, Holkin), which emitted the optimum wavelength to excite Midori Green Direct-stained DNA markers (Table 1), was applied to the DNA detection system for Midori Green Direct. The light emitted by cyan LED (490–495 nm) could not be eliminated in the reflected extra light by longpass emission filters alone (SC-52, SC-54, and SC-56) (S4B Fig, cyan LED) because the wavelength of cyan LED was near the emission wavelength. To eliminate longer wavelength region in reflected light of cyan LED, a shortpass excitation filter (510 nm) was added to the excitation cyan LED light (S4B and S4C Fig, cyan LED and cyan LED + Excitation filter, 510 nm). However, longer wavelength components in cyan-LED could not be eliminated even by a combination of a shortpass excitation filter (510 nm) and a longpass emission filter (520 nm) because cyan LED also contains longer wavelength components in the range of 490–495 nm (S4C Fig). The background noise could be effectively reduced by using a longpass emission filter (540 nm) (S4C Fig). The combination of cyan LED, shortpass excitation filter (510 nm), and longpass emission filter

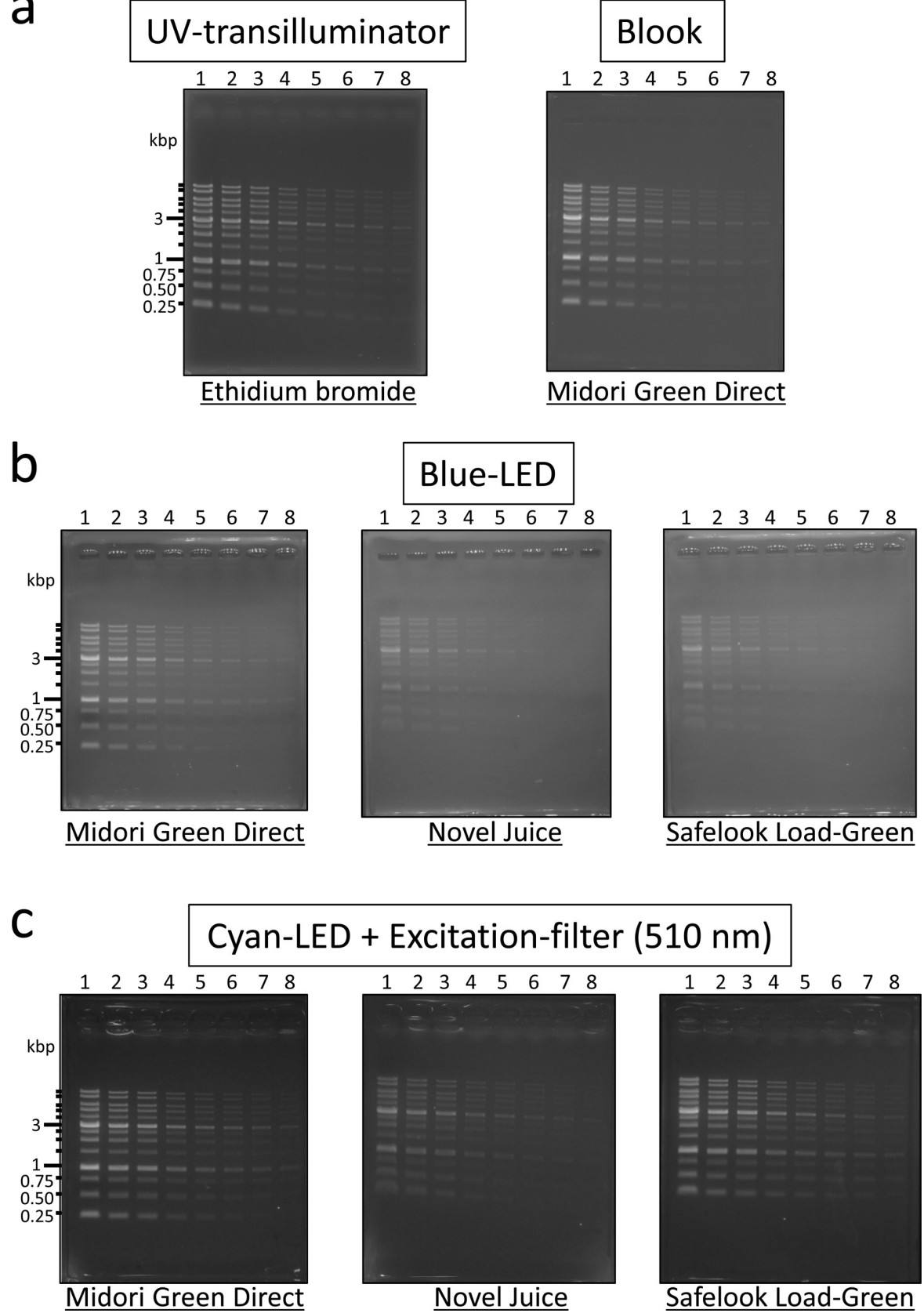

**Fig 1. Detection of DNA ladder markers stained with loading dye-type DNA-staining reagents for excitation by blue or cyan-LED light.** (a) Detection by UV (312 nm)-transilluminator system (STAGE-One, AMZ System Science) and Blook™ (470 nm, Bio-Helix); (b) Detection by blue-LED light (470 nm) excitation. SC-54 filter was used as longpass emission filter; (c) Detection by combination of cyan LED light (490–495 nm) excitation and a shortpass filter (510 nm). SC-54 filter was used as longpass emission filter. Each DNA-staining reagent was represented by underlined letters. DNA ladder markers were loaded by successive dilution. Lane 1, standard volume (5 μL (500 ng), 1 volume); lane 2, 1/2 volume; lane 3, 1/3 volume; lane 4, 1/6 volume; lane 5, 1/10 volume; lane 6, 1/15 volume; lane 7, 1/20 volume; lane 8, 1/30 volume.

(540 nm) reduced background noise and greatly improved the detection sensitivity of Midori Green Direct-stained DNA markers (Fig 1B and 1C). The detection limit of the improved cyan LED and shortpass excitation filter system was 1/10 volume of DNA markers and in the same range as that of the ethidium bromide-UV transilluminator system (Fig 1A and 1C). The combination system of cyan LED and the shortpass excitation filter could be also applied to other DNA-staining reagents, such as Novel Juice and Safelook Load-Green. The combination system could detect 1/10 volume of DNA markers using Safelook Load-Green, but only 1/3 volume of DNA markers using Novel Juice (Fig 1C). These results showed that Midori Green Direct and Safelook Load-Green detected DNA markers with higher sensitivity compared with Novel Juice. The hardware cost of the excitation system built by combining cyan LED and a shortpass filter was ~$280, lower than the cost of the commercially available Blook system (~$730). SC-54, as a longpass emission filter, effectively reduced background noise in both the blue LED (S4A Fig) and cyan LED systems combined with the excitation filter (510 nm) (S4C Fig).

## Low-cost detection system of DNA-fragments based on UV-A-excited DNA-staining reagents

The optimum excitation wavelengths for EZ-Vision and Safelook Load-White were the longer waves of UV-A (360–370 nm) (Table 1). Black light has a peak emission spectrum at 360 nm

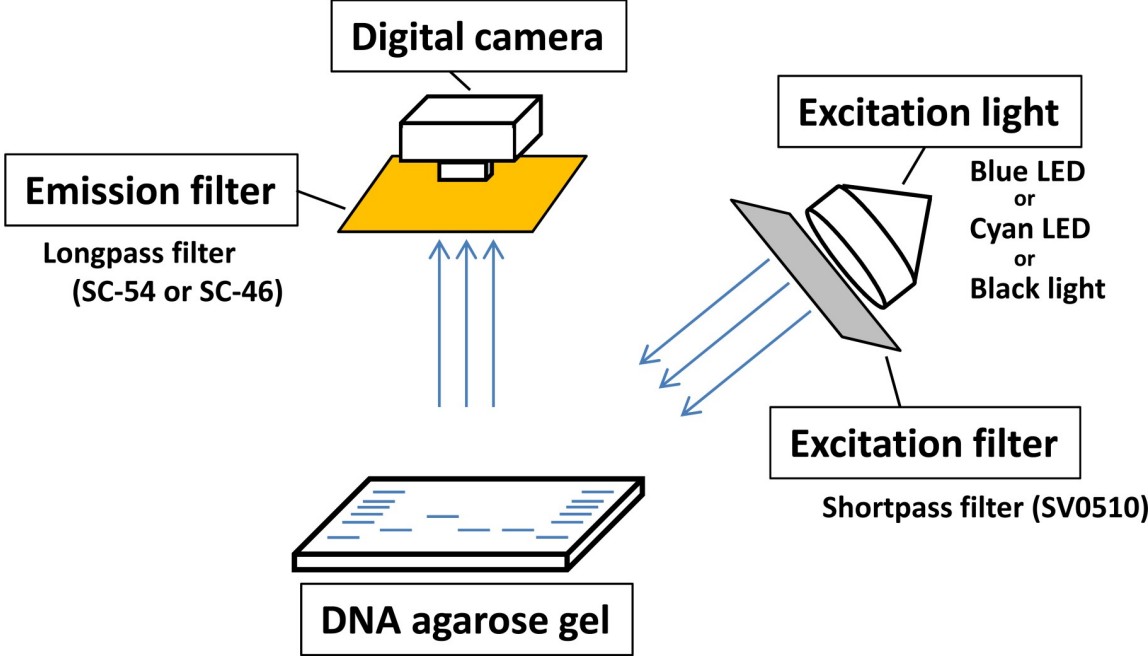

**Fig 2. Schematic diagram of DNA detection system for agarose gel electrophoresis using vertical illumination system.** Shortpass excitation filter, SV0510, was used for cyan-LED light. Longpass SC-46 emission filter was used for black light system, and longpass SC-54 emission filter was used for blue- or cyan-LED light.

and can excite DNA markers stained with EZ-Vision and Safelook Load-White. DNA markers stained with EZ-Vision and Safelook Load-White could be detected by black light, producing the 13 peaks with standard volume of DNA markers (Fig 3), whereas the commercially available DNA imaging system with UV transilluminator at 365 nm detected the DNA markers

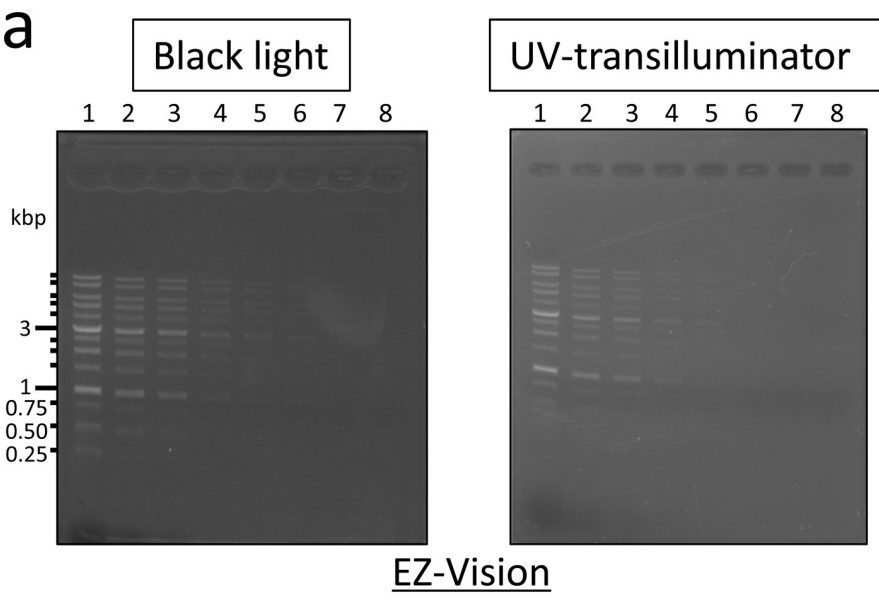

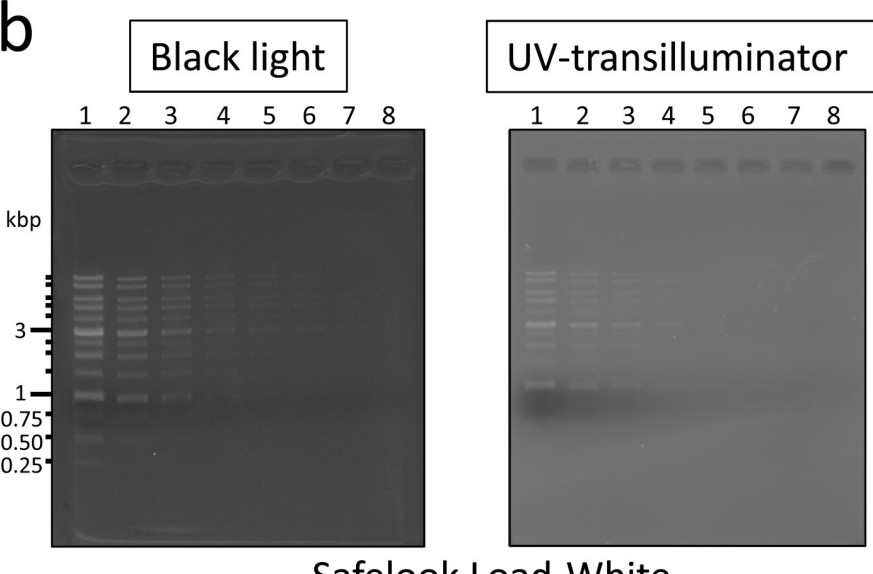

**Fig 3. Detection of DNA ladder markers stained with loading dye-type DNA-staining reagents by excitation with UV-A light.** (a) Detection of DNA fragments stained by EZ-Vision; (b) Detection of DNA fragments stained with Safelook Load-White. EZ-Vision and Safelook Load-White were excited by black light (~360 nm) or UV (365 nm) transilluminator system (STAGE-2000, AMZ System Science). Each excitation system is represented with boxed letters. SC-46 was used as longpass emission-filter for black light system, and the accessory filter of STAGE-2000 was used as an emission filter for UV (365 nm)-transilluminator system (STAGE-2000). DNA ladder markers were loaded by successive dilution. Lane 1, standard volume (5 μL (500 ng), 1 volume); lane 2, 1/2 volume; lane 3, 1/3 volume; lane 4, 1/6 volume; lane 5, 1/10 volume; lane 6, 1/15 volume; lane 7, 1/20 volume; lane 8, 1/30 volume.

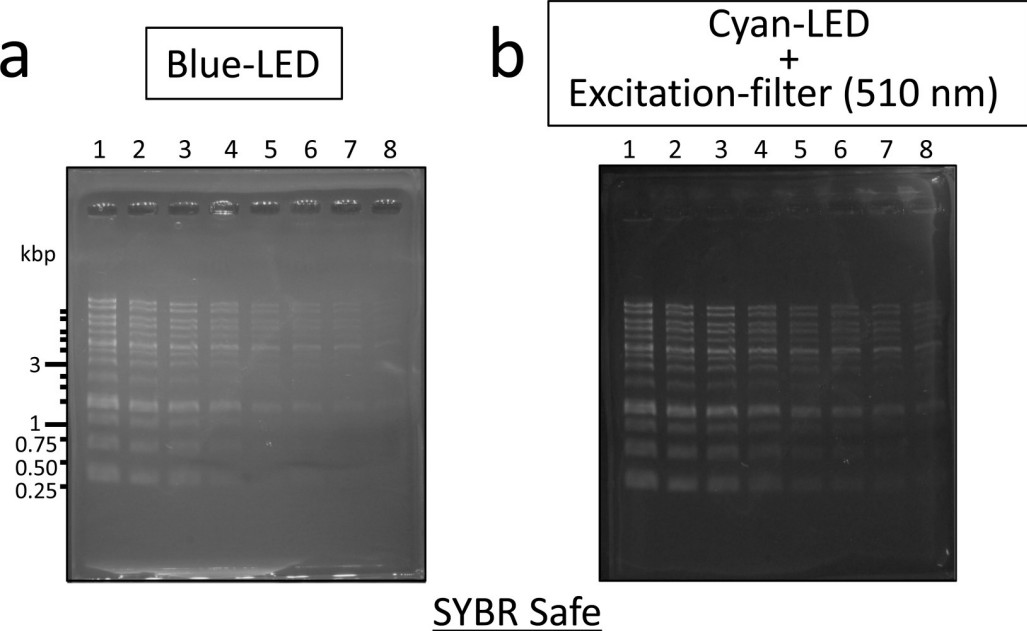

**Fig 4. Detection of DNA ladder markers stained with SYBR Safe for excitation by blue or cyan-LED light.** (a) Detection by blue-LED light (470 nm) excitation. SC-54 filter was used as longpass emission filter; (b) Detection by combination of cyan LED light (490–495 nm) excitation and a shortpass filter (510 nm). SC-54 filter was used as longpass emission filter. DNA ladder markers were loaded by successive dilution. Lane 1, standard volume (5 μL (500 ng), 1 volume); lane 2, 1/2 volume; lane 3, 1/3 volume; lane 4, 1/6 volume; lane 5, 1/10 volume; lane 6, 1/15 volume; lane 7, 1/20 volume; lane 8, 1/30 volume.

stained by EZ-Vision and Safelook Load-White with lower sensitivity. Black light is a very low-cost lamp that can be used to emit UV rays (~$40), in contrast to the expensive hardware of UV-transilluminator (>$1,000). SC-46, as a low-cost (~$16) longpass emission-filter, was effective in reducing background noise for the black light system (S5 Fig).

## Versatility of two DNA detection systems developed in this study

Both cyan-LED (or blue-LED) system and black light system can be applied to DNA gel extraction from agarose gel because agarose gel fits in the open space of both systems (Fig 2 and S3 Fig). During the cutting of the agarose gel to extract DNA fragments, DNA bands on the agarose gel excited by specific light can be visualized by orange or yellow spectacles, as an alternative to emission filters (SC-54 and SC-46). Orange spectacles (UVP, UVC-310 and Optocode, OG-HC) can be used as an alternative to SC-54 (orange filter) (S6A Fig), and yellow spectacles (TRUSCO, TSG-814Y) as an alternative to SC-46 (yellow filter) (S6B Fig).

SYBR Safe, which is widely used as a post-staining type or precast-gel type reagent, is also a non-toxic DNA-staining reagent, although its feature is different from that of the loading dye-type DNA-staining reagents used in this study. Excitation and emission wavelengths of SYBR Safe show similar features to those of Midori Green Direct, Novel Juice, and Safelook Load-Green (Table 1). To extend the versatility of the DNA detection system developed in this study, DNA agarose gel was stained with SYBR Safe (Fig 4). Consequently, DNA agarose gel stained with SYBR Safe was clearly detected by both the blue-LED and cyan-LED + Excitation filter systems. Similar to Midori Green Direct, Novel Juice, and Safelook Load-Green, SYBR Safe provided higher detection sensitivity of DNA on agarose gel by cyan-LED + Excitation

**Table 2. Sensitivities and costs of systems implemented in the detection of DNA fragments on agarose gel after electrophoresis.**

| DNA-staining reagent | Excitation system[*1] | Emission filter[*2] | Sensitivity[*3] | Hardware cost[*4] | Running cost[*5] |
|---|---|---|---|---|---|
| Ethidium bromide | UV (312 nm) transilluminator | 600FS80[*6] | High | **** | 0.05 |
| Midori Green Direct | Blue LED (Blook$^{TM}$) | Blue LED (Blook$^{TM}$) | Moderate–High | *** | 2.0 |
| | Blue LED | SC-54 | Moderate–High | ** | 2.0 |
| | Cyan LED + Ex-filter (510 nm) | SC-54 | High | ** | 2.0 |
| Novel Juice | Cyan LED + Ex-filter (510 nm) | SC-54 | Moderate–High | ** | 1.9 |
| Safelook Load-Green | Cyan LED + Ex-filter (510 nm) | SC-54 | High | ** | 3.3 |
| SYBR Safe[*7] | Blue LED | SC-54 | Moderate–High | ** | 12 or 21[*8] |
| | Cyan LED + Ex-filter (510 nm) | SC-54 | High | ** | 12 or 21[*8] |
| EZ-Vision | Black light | SC-46 | Moderate | * | 1.0 |
| | UV (365 nm) transilluminator | BP-5465[*6] | Low | **** | 1.0 |
| Safelook Load-White | Black light | SC-46 | Moderate | * | 2.0 |
| | UV (365 nm) transilluminator | BP-5465[*6] | Low | **** | 2.0 |

[*1] Excitation systems are described in detail in materials and methods. A shortpass filter (50 × 50 mm, SV0510; Asahi Spectra, Tokyo, Japan) was used as excitation filter (Ex-filter, 510 nm). Commercially available UV (312 nm)-transilluminator system (STAGE-One; AMZ System Science) was used for detection of ethidium bromide-stained DNA. Commercially available Blook$^{TM}$ (Bio-Helix) was used for detection of Midori Green Direct-stained DNA. Commercially available UV (365 nm)-transilluminator system (STAGE-2000; AMZ System Science) was used for detection of EZ-Vision- and Safelook Load-White-stained DNA.

[*2] Longpass filters SC-46 and SC-54 (Fujifilm) were used as emission filters.

[*3] Sensitivities are represented by four-grade evaluation such as low, moderate, moderate-high, and high. Detected sensitivity was defined in materials and methods.

[*4] Hardware costs containing excitation systems and emission filters are represented by * <$100, ** $100–300, *** $300–1,000, **** >$1,000.

[*5] Running costs of DNA-staining reagents are represented as the ratio to the cost of EZ-Vision, which is assumed 1.0.

[*6] These filters are accessories of UV-transilluminator systems.

[*7] SYBR Safe is widely used as a nontoxic staining reagent for DNA agarose gel staining. Although SYBR Safe is a DNA-staining reagent for post-staining or precast gel staining, it was added in Table 2 to extend the versatility of the LED-illumination system developed in this study.

[*8] Running cost of SYBR Safe was estimated as 21 for post-staining, or 12 for precast gel staining.

filter system than by blue-LED alone (Fig 4 and S7 Fig). The detection sensitivities were the same as those of Midori Green Direct and Safelook Load-Green (Figs 1 and 4).

## Conclusion

Sensitivities and costs of the DNA detection system using non-mutagenic and loading dye-type DNA-staining reagents are summarized in Table 2. Midori Green Direct and Safelook Load-Green can be used for highly sensitive DNA-detection systems on DNA agarose gel in combination with cyan LED (490–495 nm) and shortpass excitation filter (510 nm) (Fig 1C); although the running cost of Safelook Load-Green is 1.7-fold that of Midori Green Direct. SYBR Safe, a post-staining reagent, can be also applied to the LED systems (Fig 4), but its running cost is higher than that of loading dye-type DNA-staining reagents because SYBR Safe is used as a staining reagent for post-staining or precast gel staining (Table 2). The sensitivities of the DNA detection system with cyan-LED developed in this study were comparable to a commercially available ethidium bromide-UV (312 nm) transilluminator system (Fig 1A and 1C). The comparison of the initial costs for laboratory set up revealed that hardware cost of the system that combined cyan LED (490–495 nm) and shortpass excitation filter (510 nm), including emission filter, was ~$300 and thus cheaper than the cost of Blook (~$730) and UV transilluminator (>$1,000).

Another DNA-detection system was developed that was excited by black light (~360 nm). This system caused less damage compared with damage caused by short-wave UV rays (312 nm). EZ-Vision-stained and Safelook Load-White-stained DNA gels could be observed with a black light system. Running cost of EZ-Vision was half the cost of Midori Green Direct, and therefore EZ-Vision DNA staining was cost effective, although its sensitivity was less than that of Midori Green Direct. The hardware costs of EZ-Vision were <$100, and therefore the black light system is a cost-effective excitation system for DNA agarose gel electrophoresis.

Midori Green Direct should be used when high detection sensitivity is required, such as to detect small amounts of DNA fragments, whereas EZ-Vision can be implemented when DNA fragments are detected as part of the routine DNA work because EZ-Vision is the cheapest among non-mutagenic and loading dye-type DNA-staining reagents (Table 2).

## Supporting information

**S1 Fig. Detection limits by successive dilutions of DNA markers.** Profiles of the detection limit under each condition were analyzed by "Plot Profile" of Image-J. Each excitation system is represented with boxed letters, and each DNA-staining reagent is represented by underlined letters. Blue arrows indicate detectable DNA bands. (a) Fig 1A left, lane 5; (b) Fig 1A right, lane 4; (c) Fig 1B left, lane 3; (d) Fig 1C left, lane 5; (e) Fig 1V middle, lane 3; (f) Fig 1V right, lane 5; (g) Fig 3A left, lane 1; (h) Fig 3A right, lane 1; (i) Fig 3B left, lane 1; (j) Fig 3B right, lane 1.
(PPTX)

**S2 Fig. Parts of the excitation system used in this study.** (a) Blue-LED (Compact LED viewer, Sanplatec); (b) Left, cyan-LED (hlk-12led-490nm-495nm, Holkin), right, cyan-LED with excitation filter (SV0510; Asahi Spectra); (c) left, compact black light blue lamp 27 W (360 nm, FPL27BLB; Sankyo Denki), right, black light to be mounted on a fluorescent lamp stand.
(PPTX)

**S3 Fig. A photograph of DNA agarose gel electrophoresis detection systems developed in this study.** Gel images were recorded with a digital camera PowerShot G12 (Canon) in a dark place shaded by blackout curtain. (a) Blue-LED system; (b) Cyan-LED + Excitation filter system; (c) Black light system.
(PPTX)

**S4 Fig. Evaluation of emission filters for detection of DNA markers stained with Midori Green Direct.** (a) Excitation by blue-LED light (470 nm); (b) Excitation by cyan-LED light (490–495 nm); (c) Excitation by combination of cyan LED (490–495 nm) and a shortpass filter (510 nm). SC-52, SC-54, and SC-56 filters were evaluated as longpass emission-filters. DNA ladder markers were loaded by successive dilution. Lane 1, standard volume (5 µL (500 ng), 1 volume); lane 2, 1/2 volume; lane 3, 1/3 volume; lane 4, 1/6 volume; lane 5, 1/10 volume; lane 6, 1/15 volume; lane 7, 1/20 volume; lane 8, 1/30 volume.
(PPTX)

**S5 Fig. Evaluation of emission filters for detection of DNA markers stained with DNA-staining reagents excited by black light system.** (a) Detection of DNA markers stained with EZ-Vision; (b) Detection of DNA markers stained with Safelook Load-White. Black light (~360 nm) was used to excite EZ-Vision and Safelook Load-White. SC-42, SC-46, and SC-48 filters were evaluated as longpass emission filters. DNA ladder markers were loaded by successive dilution. Lane 1, standard volume (5 µL (500 ng), 1 volume); lane 2, 1/2 volume; lane 3, 1/

3 volume; lane 4, 1/6 volume; lane 5, 1/10 volume; lane 6, 1/15 volume; lane 7, 1/20 volume; lane 8, 1/30 volume.
(PPTX)

**S6 Fig. Orange and yellow spectacles used to visualize DNA while cutting it out from the agarose gel.** (a) Orange spectacles (UVP, UVC-310) can be used as an alternative to SC-54, which is an orange filter. (b) Yellow spectacles (TRUSCO, TSG-814Y) can be used as an alternative to SC-46, which is a yellow filter.
(PPTX)

**S7 Fig. Detection limits by successive dilutions of DNA markers.** Profiles of detection limit under each condition were analyzed by "Plot Profile" of Image-J. Each excitation system is represented with boxed letters, and DNA-staining reagent is represented by underlined letters. Blue arrows indicate detectable DNA bands. (a) Fig 4A , lane 3; (b) Fig 4B , lane 5.
(PPTX)

## Acknowledgments

I thank Dr. Akira Kawabe (Kyoto Sangyo University) for allowing us to use STAGE-One UV-transilluminator system (312 nm excitation). I also thank Dr. Seisuke Kimura (Kyoto Sangyo University) for allowing us to use STAGE-2000 UV-transilluminator system (365 nm excitation).

## Author Contributions

**Conceptualization:** Ken Motohashi.

**Formal analysis:** Ken Motohashi.

**Funding acquisition:** Ken Motohashi.

**Investigation:** Ken Motohashi.

**Methodology:** Ken Motohashi.

**Validation:** Ken Motohashi.

**Writing – original draft:** Ken Motohashi.

**Writing – review & editing:** Ken Motohashi.

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
