## [Decision Letter · Decision Letter 0]

31 Jul 2019

PONE-D-19-19358

Development of highly sensitive and low-cost DNA agarose gel electrophoresis detection systems and evaluation of non-mutagenic and loading dye-type DNA-staining reagents

PLOS ONE

Dear Dr. MOTOHASHI,

Thank you for submitting your manuscript to PLOS ONE. After careful consideration, we feel that it has merit but does not fully meet PLOS ONE’s publication criteria as it currently stands. Therefore, we invite you to submit a revised version of the manuscript that addresses the points raised during the review process.

We would appreciate receiving your revised manuscript by Sep 14 2019 11:59PM. To enhance the reproducibility of your results, we recommend that if applicable you deposit your laboratory protocols in protocols.io, where a protocol can be assigned its own identifier (DOI) such that it can be cited independently in the future. For instructions see: http://journals.plos.org/plosone/s/submission-guidelines#loc-laboratory-protocols

We look forward to receiving your revised manuscript.

Kind regards,

Ruslan Kalendar, PhD

Academic Editor

PLOS ONE

1. PLOS ONE now requires that authors provide the original uncropped and unadjusted images underlying all blot or gel results reported in a submission’s figures or Supporting Information files. This policy and the journal’s other requirements for blot/gel reporting and figure preparation are described in detail at https://journals.plos.org/plosone/s/figures#loc-blot-and-gel-reporting-requirements and https://journals.plos.org/plosone/s/figures#loc-preparing-figures-from-image-files. When you submit your revised manuscript, please ensure that your figures adhere fully to these guidelines and provide the original underlying images for all blot or gel data reported in your submission. See the following link for instructions on providing the original image data: https://journals.plos.org/plosone/s/figures#loc-original-images-for-blots-and-gels.

Reviewers' comments:

Reviewer's Responses to Questions

**Comments to the Author**

 5. Review Comments to the Author

Reviewer #1:

Motohashi reports the use of different illumination systems to detect DNA fragments stained with a set of six intercalating dyes and separated by agarose gel electrophoresis. The illumination systems included a conventional UV transilluminator, a Blook Blue LED system, a cyan LED system, and a blacklight transilluminator. Samples are loaded in successively lower amounts on separate lanes of the gel. Data are presented as black-and-white photographs, which were generated using a Canon Powershot camera. The images were processed using Image-J.

I believe that the article satisfies most, but not all of PLOS ONE's criteria for publication:

1. The study presents the results of primary scientific research.

2. To the best of my knowledge, the results have not been published elsewhere.

3a. Some experimental details are missing. In particular, my searchs were unsuccessful for the "Compact LED viewer [470 nm], Sanplatec, for the Cyan LED (490–495 nm, Holkin), for the SV0510 shortness filter mentioned, and for the compact blacklight lamp (FPL27BLB; Sankyo Denki) mentioned on page 4 and 6 of the manuscript. URLs would be useful.

3b. I do not understand the design of the illumination systems, and those systems are not described in sufficient detail for me to reproduce the results. Were the LEDs and black lights used in a commercial transilluminator, or were illuminators used with a homemade illuminator? A diagram of the homebuilt illuminators is required.

3c. It is striking that intercalating dyes from Molecular Probes are not included in this study. In particular, SYBR Safe is marketed as having lower toxicity than ethidium bromide and is used with high sensitivity for DNA detection in gel electrophoresis.

3d. I do not see the Image-J data used to construct Table 2. The criterion for the sensitivity values in Table 2 are arbitrary because no definition of fragment detection is given. The definition of detection limit is not standard. Statistical analyses of the data are missing.

3e. The spectrometer used to generate the excitation and emission wavelengths used in Table 1 is not described in the text. It would be useful to include the spectra in the manuscript. Alternatively, if these are literature values, the source of the information should be listed.

4. The author makes several conclusions. Some appear to be justified by the data, but one set of conclusions is not justified. The author compares the cost of a commercial transilluminator with the cost of the components used to construct the black-light illumination system. As a rule of thumb, the hardware components for a commercial instrument are ~10% of the cost of the final product; the commercial instrument has additional costs associated with design and development, regulatory approval, manufacturing, and marketing; once commercialized, the systems proposed by the author will have the similar costs as the other commercial instruments. The conclusion concerning the relative costs of the various systems must be removed from the paper.

5. Except for the missing experimental details, the manuscript is presented in an intelligible fashion. While the paper would benefit from minor copyediting, that editing is not required to the paper to be interpretable

6. The paper apparently meets ethical standards.

7. The paper presents raw data as images. The corresponding Image-J data are not presented.

Reviewer #2:

The detection system described in the paper will be of significant interest to individuals and organizations for which the cost of a conventional system is prohibitively expensive. They will be able to build their own instrument based on this paper. I recommend that the author provide more details of how the system can be constructed from commercially available parts, including an image of what it looks like, so that a person who is interested will be able to replicate it.

Reviewer #3:

This manuscript describes the development of low-cost imaging systems for detecting DNA, with the goal of minimizing DNA damage due to the illumination wavelengths. The author described the detection limits using several different DNA-binding dyes and several different illumination systems.

I have several concerns about this report:

First, the sensitivity estimation relied completely on visual perception. A more rigorous numerical method should be used. These are digital images that can be easily quantified using free software tools such as ImageJ.

Second, the author did not mention how the images were displayed and if any image contrasting or enhancement was used. Was the camera mounted inside a light-tight enclosure or setup in an open space?

Third, the lanes of all gel images in all of the figures should be labeled.

Finally, and most importantly, since the goal of this work was to minimize photo-induced DNA damage, I assume the author's ultimate goal is to excise DNA fragments from gels for cloning purposes. The author needs to demonstrate the utility of these low cost systems for visualizing and excising gel fragments. Systems that require bandpass emission filters may not enable one to look directly at the gel for manually cutting out of gel fragments. In addition, are the home-built imagers amenable to the cutting of gel fragments. Commercial transilluminators are designed to enable gel cutting.

If the author's goal is to simply document DNA gels and not enable gel cutting, then it is not clear if these low-cost alternatives offer the uniformity and repeatability of commercial imaging systems.

---

## [Author Response · Author response to Decision Letter 0]

23 Aug 2019

For Reviewer #1:

1) Some experimental details are missing. In particular, my searches were unsuccessful for the "Compact LED viewer [470 nm], Sanplatec, for the Cyan LED (490-495 nm, Holkin), for the SV0510 shortness filter mentioned, and for the compact black light lamp (FPL27BLB; Sankyo Denki) mentioned on page 4 and 6 of the manuscript. URLs would be useful.

OUR ANSWER:

Thank you for your suggestion. Information of blue- and cyan-LED system, black light system, and shortpass filter (SV0510) was added in materials and methods (p4, l　98-106).

2) I do not understand the design of the illumination systems, and those systems are not described in sufficient detail for me to reproduce the results. Were the LEDs and black lights used in a commercial transilluminator, or were illuminators used with a homemade illuminator? A diagram of the homebuilt illuminators is required.

OUR ANSWER:

Schematic diagram of these DNA detection systems was added as Figure 2. Moreover, photographs of several parts to build the DNA detection systems were displayed as Figure S2. In addition, several sentences were added in text (p6, l 153-157).

3) It is striking that intercalating dyes from Molecular Probes are not included in this study. In particular, SYBR Safe is marketed as having lower toxicity than ethidium bromide and is used with high sensitivity for DNA detection in gel electrophoresis.

OUR ANSWER:

Thank you for your suggestion. In this study, I developed two DNA detection system using loading dye-type DNA-staining reagents, NOT post-staining type reagents such as SYBR safe. However, SYBR safe is widely used. Therefore, I evaluated detection sensitivities of DNA agarose gel stained with SYBR safe (Figure 4, Figure S7, p2 l 29-30, p8 l 220-232, and p8 l 240-243). 

4) I do not see the Image-J data used to construct Table 2. The criterion for the sensitivity values in Table 2 are arbitrary because no definition of fragment detection is given. The definition of detection limit is not standard. Statistical analyses of the data are missing.

OUR ANSWER:

Thank you for your comments. The Image-J data were shown in Figure S1 and S7.

5) The spectrometer used to generate the excitation and emission wavelengths used in Table 1 is not described in the text. It would be useful to include the spectra in the manuscript. Alternatively, if these are literature values, the source of the information should be listed.

OUR ANSWER:

Thank you for your comment. Excitation and emission wave lengths were referred to each instruction manual. The source of excitation and emission wave lengths was described in Table 1 legend.

6) The author makes several conclusions. Some appear to be justified by the data, but one set of conclusions is not justified. The author compares the cost of a commercial trans-illuminator with the cost of the components used to construct the black-light illumination system. As a rule of thumb, the hardware components for a commercial instrument are ~10% of the cost of the final product; the commercial instrument has additional costs associated with design and development, regulatory approval, manufacturing, and marketing; once commercialized, the systems proposed by the author will have the similar costs as the other commercial instruments. The conclusion concerning the relative costs of the various systems must be removed from the paper.

OUR ANSWER:

Thank you for your comments. Costs of commercially available hardware include design and development, regulatory approval, and manufacturing etc., as described by reviewer #1. Therefore, commercially available systems are prohibitively expensive, as pointed out by reviewer #2. In this study, I would like to compare initial cost for laboratory setting up. These handmade LED-systems could be set up by 1/2 to 1/3, compared with commercially available Blook system or UV-transilluminator system. Furthermore, Black light system could be set up by ~$100. I believe that cost down of these instruments is helpful for researchers in a laboratory. My final goal is not that these developed DNA detection systems is marketed, and an initial costs for laboratory set up is reduced. For this purpose, hardware cost was described in text, however, I edited this sentence to explain these contexts (p8, l 246-249). I believe that comparison of initial hardware costs between commercially system and the developed systems in this study is significant information to build their DNA detection system in a laboratory.

7) Except for the missing experimental details, the manuscript is presented in an intelligible fashion. While the paper would benefit from minor copyediting, that editing is not required to the paper to be interpretable.

OUR ANSWER:

Several figures and supplemental figures were added to explain experimental details (Figure 2 and 4, Figure S1, S2, S3, S6, and S7).

For Reviewer #2:

1) The detection system described in the paper will be of significant interest to individuals and organizations for which the cost of a conventional system is prohibitively expensive. They will be able to build their own instrument based on this paper. I recommend that the author provide more details of how the system can be constructed from commercially available parts, including an image of what it looks like, so that a person who is interested will be able to replicate it.

OUR ANSWER:

Thank you for your valuable suggestion, schematic diagram of these systems and photographs of several parts were added as Figure 2, Figure S2, S3, and S6, to build their own instruments based on this study. In addition, several sentences in text were modified and added (p5 l 124-125, p6 l 153-157, and p7 l 211-219)

 

For Reviewer #3:

1) The sensitivity estimation relied completely on visual perception. A more rigorous numerical method should be used. These are digital images that can be easily quantified using free software tools such as ImageJ.

OUR ANSWER:

Thank you for your valuable comments, data in this article were analyzed by “Plot Profile” of Image-J, as described in materials and methods (p5, l 128-135). To demonstrate clearly, data analyzed with Image-J were displayed as Figure S1 and S7.

2) The author did not mention how the images were displayed and if any image contrasting or enhancement was used. Was the camera mounted inside a light-tight enclosure or setup in an open space?

OUR ANSWER:

All gel images were not applied any image processing. It was mentioned in text as “None of images were not altered by image processing methods such as contrasting and enhancement.” (p5, l 125-126). Schematic diagram of developed DNA detection systems was displayed in Figure 2. Two DNA detection systems were set up in blackout curtain (Figure S3). However, if light switches can be shot down in its room, the systems can be set up in an open space.

3) The lanes of all gel images in all of the figures should be labeled.

OUR ANSWER:

Thank you for your suggestion. The lanes of all gel images were labeled.

4) Since the goal of this work was to minimize photo-induced DNA damage, I assume the author's ultimate goal is to excise DNA fragments from gels for cloning purposes. The author needs to demonstrate the utility of these low cost systems for visualizing and excising gel fragments. Systems that require bandpass emission filters may not enable one to look directly at the gel for manually cutting out of gel fragments. In addition, are the home-built imagers amenable to the cutting of gel fragments? Commercial trans-illuminators are designed to enable gel cutting.

If the author's goal is to simply document DNA gels and not enable gel cutting, then it is not clear if these low-cost alternatives offer the uniformity and repeatability of commercial imaging systems.

OUR ANSWER:

Thank you for your valuable suggestion. Excitation and emission filters do not disturb accessing to agarose gel for gel-cutting. Two developed DNA detection system can be applied to gel-cutting for DNA extraction from agarose gel, and the schematic diagram and photographs were shown (Figure 2 and Figure S3). In addition, orange or yellow spectacles can be used as emission-filters for gel-cutting (Figure S6). These facts were described in text (p7, l 211-219).

---

## [Editor Report · Decision Letter 1]

26 Aug 2019

Development of highly sensitive and low-cost DNA agarose gel electrophoresis detection systems, and evaluation of non-mutagenic and loading dye-type DNA-staining reagents

PONE-D-19-19358R1

Dear Dr. MOTOHASHI,

We are pleased to inform you that your manuscript has been judged scientifically suitable for publication and will be formally accepted for publication once it complies with all outstanding technical requirements.

With kind regards,

Ruslan Kalendar, PhD

Academic Editor

PLOS ONE

---

## [Editor Report · Acceptance letter]

28 Aug 2019

PONE-D-19-19358R1 

Development of highly sensitive and low-cost DNA agarose gel electrophoresis detection systems, and evaluation of non-mutagenic and loading dye-type DNA-staining reagents 

Dear Dr. MOTOHASHI:

I am pleased to inform you that your manuscript has been deemed suitable for publication in PLOS ONE. Congratulations! Your manuscript is now with our production department. 

With kind regards,

on behalf of

Dr. Ruslan Kalendar 

Academic Editor

PLOS ONE